# Multi-orbital charge transfer at highly oriented organic/metal interfaces

Giovanni Zamborlini [1], Daniel Lüftner [2], Zhijing Feng [3,4], Bernd Kollmann[2], Peter Puschnig [2], Carlo Dri [3,4], Mirko Panighel[5], Giovanni Di Santo [5], Andrea Goldoni[5], Giovanni Comelli[3,4], Matteo Jugovac[1], Vitaliy Feyer[1] & Claus Michael Schneider[1,6]

The molecule–substrate interaction plays a key role in charge injection organic-based devices. Charge transfer at molecule–metal interfaces strongly affects the overall physical and magnetic properties of the system, and ultimately the device performance. Here, we report theoretical and experimental evidence of a pronounced charge transfer involving nickel tetraphenyl porphyrin molecules adsorbed on Cu(100). The exceptional charge transfer leads to filling of the higher unoccupied orbitals up to LUMO+3. As a consequence of this strong interaction with the substrate, the porphyrin's macrocycle sits very close to the surface, forcing the phenyl ligands to bend upwards. Due to this adsorption configuration, scanning tunneling microscopy cannot reliably probe the states related to the macrocycle. We demonstrate that photoemission tomography can instead access the Ni-TPP macrocycle electronic states and determine the reordering and filling of the LUMOs upon adsorption, thereby confirming the remarkable charge transfer predicted by density functional theory calculations.

[1] Peter Grünberg Institute (PGI-6), Forschungszentrum Jülich GmbH, D-52425 Jülich, Germany. [2] Institut für Physik, Karl-Franzens-Universität Graz, NAWI Graz, 8010 Graz, Austria. [3] Department of Physics, University of Trieste, Via A. Valerio 2, 34127 Trieste, Italy. [4] IOM-CNR Laboratorio TASC, S.S. 14 km 163.5 in AREA Science Park, Basovizza, I-34149 Trieste, Italy. [5] Elettra—Sincrotrone Trieste, S.S. 14 km 163.5 in AREA Science Park, Basovizza, I-34149 Trieste, Italy. [6] Fakultät f. Physik and Center for Nanointegration Duisburg-Essen (CENIDE), Universität Duisburg-Essen, D-47048 Duisburg, Germany. Correspondence and requests for materials should be addressed to G.Z. (email: g.zamborlini@fz-juelich.de) or to V.F. (email: v.feyer@fz-juelich.de)

**P**orphyrins are extremely versatile molecules, which allow for tailoring a variety of electronic, magnetic, and conformational properties[1]. These properties form the basis for several promising organic-based technological applications such as colorimetric gas sensors[2], organic spin valves[3], field-effect transistors, light-emitting diodes, optical switches[4], non-volatile data storage systems[5], and solar cells[6]. In particular, supramolecular multiporphyrin arrays are considered as functional components in nanodevices[7]. The molecule–substrate and the molecule–molecule interactions often result in charge transfer between the substrate and the lowest unoccupied and highest occupied molecular orbitals (LUMO and HOMO, respectively)[8, 9] and possibly, in case of magnetic substrates, introduce spin degrees of freedom[10, 11] via the formation of spin-polarized hybrid interface states[12, 13]. From this perspective, detailed information about the changes in the electronic structure of molecules upon adsorption at the interface is crucial to design and prototype new devices based on organic compounds[14]. Despite considerable progress, the full potential of these organic molecules is far from being exploited. A main obstacle in this respect is based on the difficulty in predicting, modeling, and measuring the electronic and structural properties of the interfaces[15].

Intense experimental and theoretical efforts have targeted the understanding of the behavior of porphyrin assemblies on metal surfaces[16, 17]. Powerful area-averaging spectroscopic techniques, such as ultraviolet and X-ray photoemission spectroscopy (UPS and XPS) or near-edge X-ray absorption fine structure (NEXAFS) were employed to obtain information on interaction sites[9] and bonding geometries[18], as well as the electronic structure[19] and magnetic properties[20] of adsorbed porphyrin molecular layers. However, these spectroscopic techniques have difficulties to unambiguously identify, and properly describe, the features related to the molecule–metal interaction. For example, interpretation of UPS and NEXAFS spectra, based on the comparison between submonolayer and thick molecular films, might be misleading. In UPS, the observed structures in the photoemission spectra may arise from purely kinematic factors, such as umklapp scattering[21], rather than molecule–metal hybridization. In NEXAFS, the full occupation of the LUMO leads to the quenching of the related resonances[22, 23]. However, the strong polarization dependence of the NEXAFS intensity and charge rearrangement in the molecule, which takes place upon adsorption, may complicate the direct determination of the molecular states involved in the charge transfer process and their energy alignment. Even the calculated density of states (DOS) can be inaccurate because of the inherent approximations of the theoretical frameworks[24]. For this reason, an experimental technique, providing direct access to the electronic structure of adsorbed molecules, would be desirable to experimentally shed light on the charge redistribution at the molecule–metal interface. In the present work, we unequivocally determine the electronic occupancy of the molecular frontier orbitals, i.e., HOMOs and LUMOs, of the adsorbed nickel tetraphenyl porphyrin (Ni-TPP) on the Cu(100) surface by means of photoemission tomography (PT). Our findings indicate a pronounced electron charge transfer from the Cu(100) surface to the macrocycle of the Ni-TPP molecules, involving even the gas-phase LUMO + 3 of Ni-TPP. By comparing scanning tunneling microscopy (STM) and density functional theory (DFT)-simulated images, we demonstrate that the topography contrast arises mainly from the electronic states of the porphyrin phenyl rings, which are strongly tilted upwards. This molecular configuration allows the macrocycle to get close to the copper surface, while preventing the frontier orbitals, spatially located on this moiety, to be probed by STM.

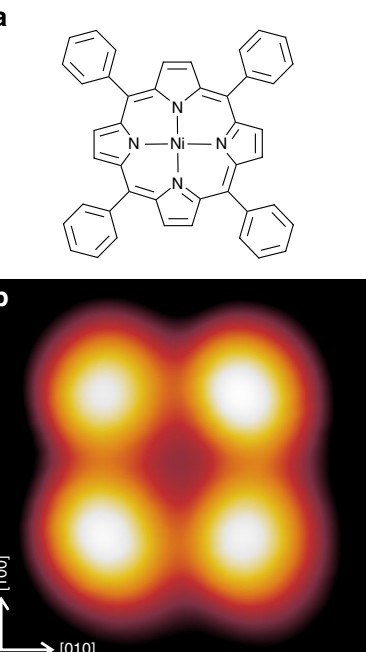

**Fig. 1** Ni-TPP molecule. **a** Chemical structure of Ni-TPP. **b** STM image of a single isolated Ni-TPP on Cu(100). STM image parameters: $V_b = -0.2\,V$, $I_t = 0.2\,nA$, $2 \times 2\,nm^2$

Combining the results from PT, STM, and DFT calculations, we develop a consistent picture of the structural and electronic properties of the Ni-TPP/Cu(100) interface. We believe that, this approach, applied to similar systems of $\pi$-conjugated molecules, may help to unravel the electronic structure of the molecular/metal interface, even in the presence of strong hybridization, ultimately improving the organic-based device-engineering.

## Results

**Geometric structure.** First, we focus on the appearance of a single Ni-TPP molecule on Cu(100) in STM. In Fig. 1, the chemical structure of Ni-TPP (a) and an STM image of the isolated molecule (b) are shown. In the STM image the four symmetric bright lobes are associated to the four phenyl terminations and the central depression to the macrocycle, with the metal in the center (see Fig. 1b). A single isolated Ni-TPP molecule is oriented along the [001] direction of Cu(100), i.e., with the N–Ni–N axes parallel to the [001] and [010] directions, respectively.

When increasing the coverage, we observe that the molecules tend to assemble into compact islands. STM images reveal the presence of two rotational domains (labeled A and B in Fig. 2a), rotated by about ±8° with respect to the [001] direction of the Cu (100) substrate. Only these two domains, mirrored with respect to the [001] direction, are observed as a consequence of the fourfold symmetry of both substrate and Ni-TPP. They are both commensurate with the substrate, and their square unit cells can be described by the following epitaxial matrices:

$$A = \begin{bmatrix} 4 & 3 \\ -3 & 4 \end{bmatrix} \quad B = \begin{bmatrix} 3 & 4 \\ -4 & 3 \end{bmatrix} \quad (1)$$

The unit cell size is determined experimentally as the distance between two molecular centers (~1.27 nm). The excellent agreement between the measured and simulated low-energy

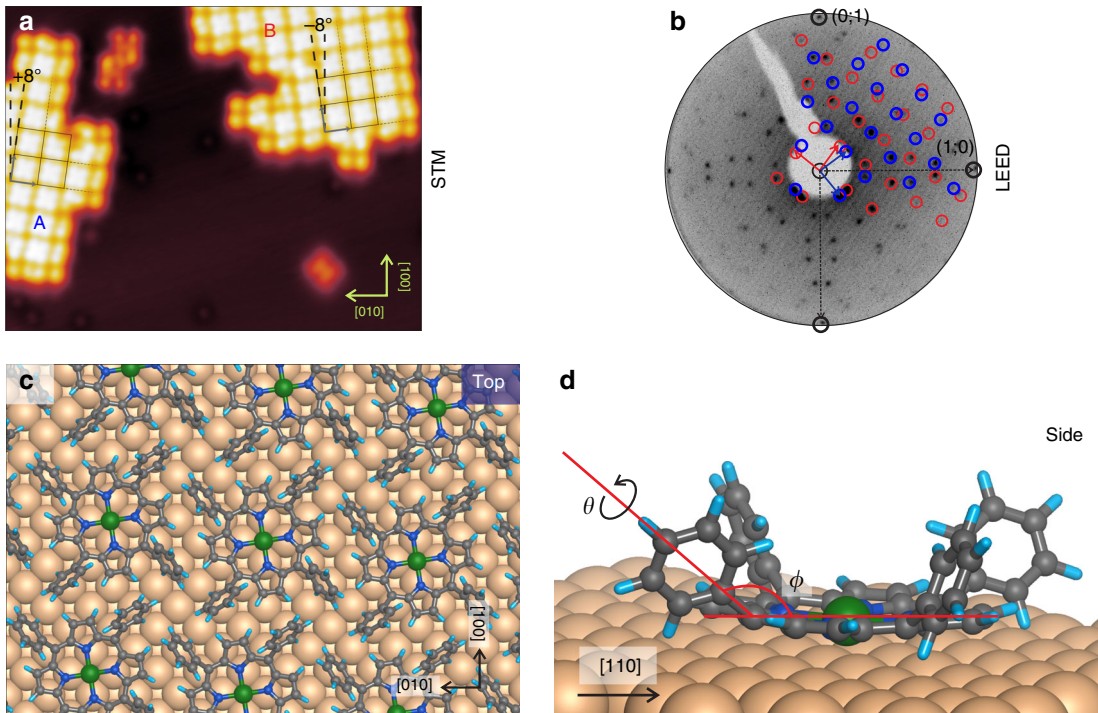

**Fig. 2** Adsorption geometry of close-packed Ni-TPPs. **a** STM image including two Ni-TPP domains, labeled with A and B, respectively. STM image parameters: $V_b = -1.5$ V, $I_t = 0.2$ nA, image size $15 \times 20$ nm$^2$, measured at 4.3 K. **b** LEED pattern of Ni-TPP/Cu(100) acquired at $E_K = 45$ eV. The simulated structure is superimposed: *blue* and *red spots* correspond to A and B domains, respectively. Proposed adsorption model for Ni-TPP/Cu(100): **c** top view and **d** side view

electron diffraction (LEED) patterns of the organic film (see Fig. 2b), based on the matrices determined above, confirms the STM findings, also indicating the existence of long-range order across the surface. Additionally, uncommon features, having a different appearance with respect to the single and close-packed Ni-TPPs, are also observed in Fig. 2a. These features can be related to species/defects on the surface, such as chemically modified rectangular-Ni-TPP or H$_2$-TPP, for example[25]. A quantitative analysis of the C 1s and N 1s core level spectra (not shown) reveals that the peak intensity ratios well agree with the Ni-TPP stoichiometry. Furthermore, no additional features have been detected in the spectra, suggesting that the amount of these species/defects on the Ni-TPP layer is below the XPS sensitivity.

In order to obtain further insight into the bonding and conformation of the adsorbed Ni-TPP molecule on Cu(100), we used the unit cell determined above to perform periodic DFT calculations of the system. For each of the three adsorption sites considered, where the central Ni atom is either at a top, bridge, or hollow site of the Cu(100) surface, we have optimized the geometry, using the vdW-surf method[26, 27] to account for the van der Waals interactions. We found the hollow site to be energetically more favorable compared to the bridge and on-top sites by 1 and 2 eV, respectively. The reason for these significant differences lies in the fact that, in the relaxed structure of the hollow site, the macrocycle can move very close to the surface (about 2.0 Å), maximizing the interactions with the Cu substrate. This has a significant impact on the molecular adsorption geometry. As already found by Donovan et al.[28] by means of combined STM experiments and DFT calculations for Co-TPP on Cu(110), the strength of the substrate–macrocycle bond has two important effects: first, the macrocycle remains almost planar in order to maximize the contact area with the substrate, and, second, significant tilt and twist deformations are induced in two distinct pairs of diametrically opposite phenyls.

Following the notation of Wölfle et al.[29], we define the tilt angle, $\phi$, as the angle between the plane of the macrocycle and the single C–C bond connecting phenyl rings to the *meso*-carbon bridge. $\theta$, instead, is the twist angle, i.e., the azimuthal rotation of the substituent around the same C-C bond (see Fig. 2d). At a twist angle of $\theta = 90°$ the phenyl groups are perpendicular to the plane defined by the porphyrin macrocycle. Our DFT simulations reveal geometrical trends similar to those observed in the case of Co-TPP on Cu(110)[28], even though in our case the geometrical distortions are slightly more pronounced. As in ref. [28], the molecule ends up very close to the surface, with the macrocycle just slightly corrugated, but with the phenyl rings pointing away from the surface under a tilt angle $\phi$ of ~140°, and a twist angle $\theta$ of ~70° (Fig. 2c, *top*). As already analyzed in detail in the literature[28], the distortion of the molecule costs energy, which is compensated by the energy gained through the interaction with the substrate. This geometry has a strong impact on the electronic structure at the interface, as discussed below. It is also worth mentioning that during our procedure for the geometry relaxation, we found two local energy minima for each adsorption site: one where the macrocycle is located ~4 Å above the surface and the other, significantly lower in energy, with the macrocycle much closer to the surface (2 Å). In the former minimum, the phenyl ligands assume an almost gas-phase conformation ($\theta = 70°$, $\phi = 180°$), whereas in the latter they are strongly tilted, as described above. In the optimized structure, Ni-TPPs rotate by 8° with respect to the [001] direction, i.e., the N–Ni–N axes is 8° off with respect to the [001] and [010] directions.

Looking now at the STM image of the self-assembled Ni-TPP, we note that the four lobes (already assigned above to the phenyl rings) appear now asymmetric, revealing a chiral character (see Fig. 3a). The chirality arises from the twisting of the phenyls as predicted by the DFT and confirms that phenyls are no longer perpendicular to the surface, like for the isolated molecule. It has to be noted that the orientation of the Ni-TPP with respect to the

Experiment

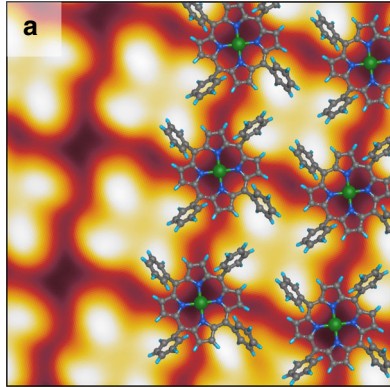

Simulation

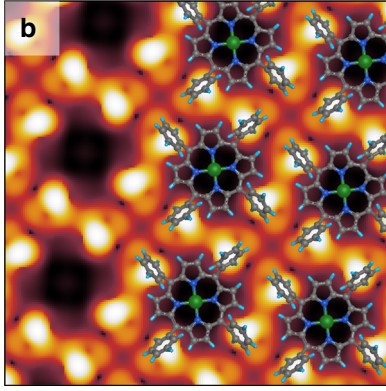

**Fig. 3** STM measured and simulated images. **a** STM image on Ni-TPP/Cu (100) film. Ni-TPP molecule is superimposed. STM image parameters: $V_b$ = + 1.5 V, $I_t$ = 0.5 nA, image size 3 × 3 nm$^2$. **b** DFT simulated STM image using the relaxed structure shown in Fig. 2

substrate can only be determined with a large uncertainty. In fact, no features related to the macrocycle are visible in the experimental images, and therefore a measurement of the azimuthal molecule–substrate angle can only rely on the protrusions related to the phenyl rings. The latter, however, appear somehow distorted due to their chiral character, making it difficult to correlate their orientation and mutual alignment with the orientation of the macrocycle. Taking into account all these sources of error, we can only state that the azimuthal molecule–substrate angle lies between ~−6° and ~10°. For this reason, a complementary experimental technique, $\mu$-ARPES, has been used to obtain this missing information, and the result will be discussed below.

Starting from the optimized adsorption structure, we simulated the STM image, and the result is shown in Fig. 3b. As already reported in refs. [28, 30], the four lobe appearance of Ni-TPP in the experimental STM images is associated with strongly tilted phenyl ring terminations, while the central dark depression is associated to the macrocycle that approaches very close to the substrate. The contrast, in both images, is vastly dominated by the features arising from the phenyl rings: due to their very large twist and tilt angles, they extend well above the plane of the macrocycle, preventing the macrocycle from being resolved by the STM tip.

Overall, the experimental and simulated images are in reasonable agreement. The domain orientation and the periodicity of the Ni-TPP self-assembled monolayer match perfectly. However, the simulated images unavoidably show

many features that the STM tip cannot resolve experimentally, due to the broadening caused by the intrinsically large width and the electronic properties of the real tip apex (in the calculations, the apex-copper surface distance is ~5.5–7.5 Å). Therefore, the phenyls in the simulation appear slightly different from the experimental image. In accordance with the simulated images, the appearance of Ni-TPP in the STM images is almost bias-independent within the [−2.0, +2.0] V range (see also the Supplementary Note 1).

**Electronic structure**. We now focus on the electronic structure of the Ni-TPP/Cu(100) interface where our main interest lies in the frontier orbitals localized at the porphyrin core. Figure 4a shows the angle-integrated photoelectron spectra of the clean Cu (100) substrate (red line) and of the Ni-TPP/Cu(100) interface (blue line). The valence band of the clean copper is dominated by the $sp$ band and has a rather featureless plateau, while the Ni-TPP/Cu(100) spectrum shows two prominent features at binding energies (BEs) of 0.15 and 0.98 eV. A third feature, at 1.73 eV, becomes resolvable only after performing $\mu$-ARPES measurements because it is hidden under the Cu 3$d$ band emissions (all three features are marked in Fig. 4a).

To clarify the origin of these features, we have computed the density of states of the Ni-TPP/Cu(100) interface by means of DFT for the optimized adsorption geometry. The DOS was calculated using the Heyd-Scuseria-Ernzerhof (HSE) short-range hybrid functional[31] to account for the presence of both the delocalized $\pi$-orbitals of the macrocycle and the localized states at the Ni atom. Note that using standard generalized gradient approximation (GGA) functionals in the calculations would lead to an incorrect orbital ordering due to problems related to the self-interaction error in DFT[24, 32]. Figure 4b shows the HSE-DOS projected onto specific molecular orbitals (PDOS). We observe that the degenerate gas-phase LUMO/LUMO + 1 (blue and light blue lines in Fig. 4b) are below the Fermi level, indicating that, upon adsorption, they become occupied. Surprisingly the gas-phase LUMO + 2 and LUMO + 3 also have a non-zero contribution below $E_F$. These molecular levels are filled by charge transfer taking place from the substrate to the molecular film.

It should be emphasized, however, that the occupancy of LUMO/LUMO + 1 and the partial occupation of LUMO + 2 and LUMO + 3 does not express the net charge transfer. As is well known from other systems[8], there is also a back donation from the molecule to the substrate. We attribute the filling of formerly unoccupied MOs, i.e., the LUMOs, to the extremely short distance between the molecular backbone and the substrate. As a further consequence of this short distance, the charge spill-out at the surface is pushed into the metal (push-back effect)[33–35]. This results in a dipole that substantially reduces the work function, thereby shifting the molecule's LUMOs below $E_F$. On the other hand, the charge transfer to the molecule creates a dipole of opposite sign. In accordance with the experimentally observed reduction of the work function upon the formation of the Ni-TPP layer, the DFT calculations suggest that for the overall interface dipole, the push-back effect dominates over charge transfer into the molecule (for further details see Supplementary Note 2).

Upon adsorption, the frontier orbitals, i.e., HOMOs and LUMOs close to the band gap, spread over a wide energy range, [−3.0; +1.5] eV (see Fig. 4b). To address the question at which parts within the molecule, these orbitals are located, we compare, in Fig. 4c, the DOS projected onto the macrocycle (green curve) with that projected onto the phenyls (light blue curve). We note that, in this energy range, the PDOS of the phenyl rings has

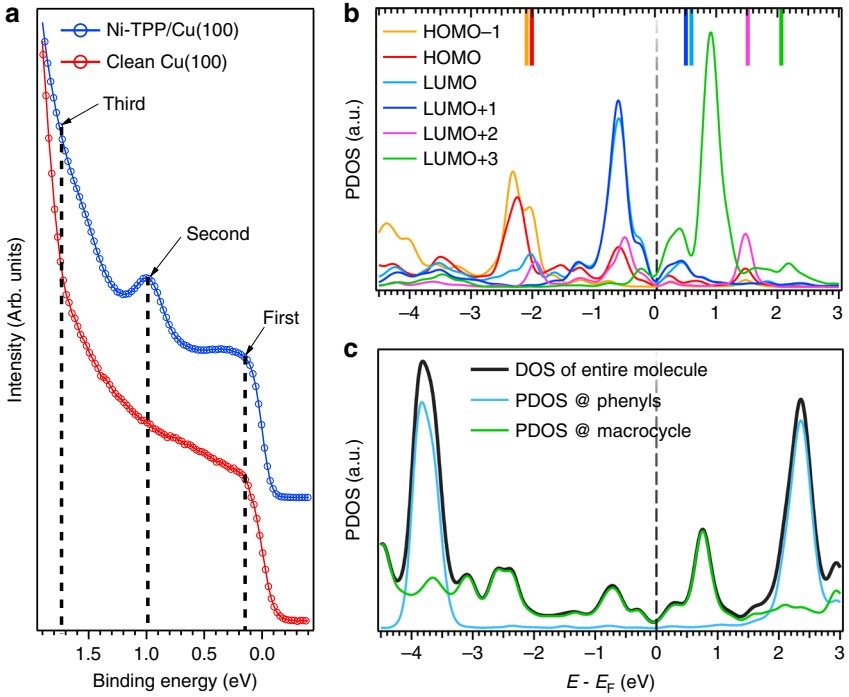

**Fig. 4** Electronic structure of Ni-TPP/Cu(100). **a** Photoemission spectra of clean Cu(100) and Ni-TPP/Cu(100) acquired at 26 eV photon energy. **b** PDOS onto molecular orbitals for the Ni-TPP/Cu(100) system. The energy position of the corresponding gas-phase molecular orbitals, aligned with respect to the vacuum level, is indicated with *colored bars* on the *top axis*. **c** DOS of the entire molecule (*black curve*), PDOS onto the phenyl group (*light blue curve*) and on the macrocycle (*green curve*)

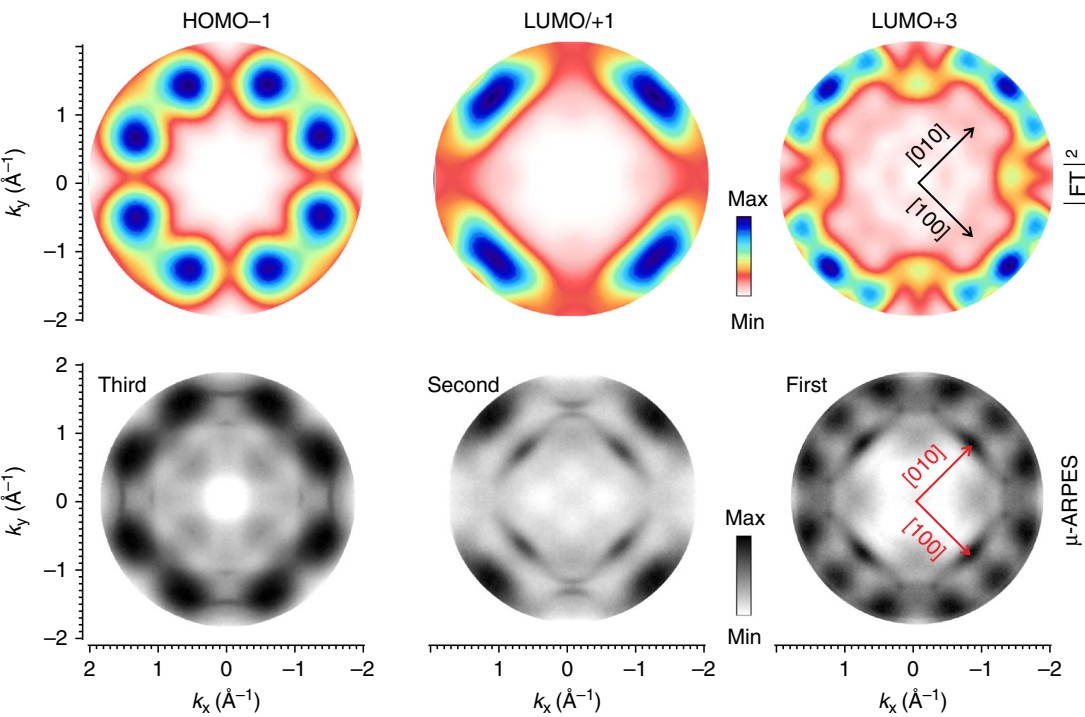

**Fig. 5** Photoemission tomography. Comparison between $\mu$-ARPES measured patterns (*bottom*) and the correspondent calculated $|FT|^2$ of the molecular orbitals (*top*). In the experimental maps, the sharp inner features are related to the *sp* band of the copper surface

a rather featureless plateau, whereas the macrocycle PDOS is structured. We can conclude that the frontier orbitals are almost entirely located on the porphyrin macrocycle. The lineshape of the phenyl PDOS, with the two strong peaks at −3.8 and +2.4 eV,

follows the contrast changes of the four lobes in the bias-dependent STM image series (see Supplementary Note 1).

The assignment of individual molecular orbitals to the observed peaks in the integrated valence band spectrum carried

out only on the basis of the calculated PDOS is problematic, due to fundamental limitations of DFT, e.g., the wrong asymptotic behavior of the potential or the band gap problem due to the derivative-discontinuity issue[36, 37]. Indeed, although the HSE functional presents some improvement over GGA, it still may lead to an incorrect energy alignment[38]. Moreover, the PDOS for various molecular states extends over several eV due to hybridization with the substrate. Therefore, a complementary experimental technique allowing for a deconvolution of the molecular emissions into contributions from different molecular orbitals would be desirable. This can be realized by utilizing the angular distribution of the photoemitted electrons in the $\mu$-ARPES experiments. In our set-up such two-dimensional momentum maps are acquired in one single shot experiment, with a wide reciprocal space range: $k_x$, $k_y \in [-2, +2]$ Å$^{-1}$. The corresponding momentum maps, measured at the BEs of the three peaks indicated in Fig. 4a, are presented in the bottom row of Fig. 5. These three maps clearly show characteristic and well-distinguishable patterns. Note that the experimental $\mu$-ARPES data, in addition to emission from Ni-TPP states, also contain $sp$ band emissions from the Cu substrate[39].

These experimental momentum maps can be compared to simulated orbital patterns based on DFT calculations. Within the PT approach[40–43], the photoemitted electron is approximated by a plane wave. Thus, the angular dependence of the photoemission intensity is determined by the Fourier transform (FT) of the respective initial state wave function. Real-space molecular orbitals and corresponding |FT|$^2$ are shown in Supplementary Note 3. Because $\mu$-ARPES averages over a large number of molecules, the two mirror domains of Ni-TPP with different azimuthal orientations of the molecules must be taken into account. Moreover, due to the degeneracy of some molecular states in the gas phase, e.g., LUMO and LUMO + 1, the simulated ARPES maps should be computed as a superposition of such degenerate states. The simulated maps are shown in the upper row of Fig. 5 and can be directly compared to the measured patterns in the lower row of the same panel. Based on the excellent agreement between experimental and theoretical momentum maps, the features in the photoelectron spectra of Ni-TPP/Cu(100) can now be unambiguously assigned to the emissions from the LUMO + 3, LUMO/LUMO + 1, and HOMO - 1 of Ni-TPP, respectively. This provides direct feedback to the theory on the choice of the best approximation for exchange-correlation effects in DFT.

Note that, the FTs, shown in Fig. 5 and in the Supplementary Note 3, were calculated by using the Ni-TPP geometry resulting upon adsorption. Computing the FTs starting from the Ni-TPP gas-phase geometry produces only small changes. This is not surprising because the corresponding states are located on the macrocycle, which is only slightly deformed upon adsorption. None of the states localized on the phenyl rings are found in the considered energy window (see Fig. 4c).

The PT analysis also confirms the tentative assignment suggested by the computed PDOS and proves that indeed the gas-phase LUMO + 3 and the degenerate LUMO and LUMO + 1 become occupied upon adsorption of Ni-TPP on the metal surface, demonstrating the strong chemisorptive nature of the Ni-TPP/Cu interaction. The results also show that the HOMO - 1, rather than the HOMO, is the origin of the feature at a BE of 1.73 eV. Note that the HOMO and HOMO - 1 are quite close in energy for the gas-phase Ni-TPP and the PDOS of the Ni-TPP/Cu interface also predicts that the HOMO and HOMO - 1 change their energetic order. Since the HOMO - 1 peak sits on the onset of the strong Cu $d$-band emissions, no molecular features were observed below that energy range. The partial occupation of the gas-phase LUMO + 2 suggested by the PDOS result is not confirmed by the ARPES momentum maps. Note that this might be related to the fact that the LUMO + 2 momentum map shows only four narrow lobes, presumably located outside the experimentally probed k-space range (see Supplementary Note 3).

Exploiting the clear signals from the $sp$ band of the Cu substrate in the momentum maps, the orientation of the molecule with respect to the substrate high-symmetry directions can be easily determined. The best agreement between all experimental and corresponding simulated momentum maps is for an azimuthal orientation of Ni-TPP of $\pm 8°$ with respect to the [001] direction of the substrate. Within experimental error bars, this agrees with the orientation determined by the analysis of the STM images.

## Discussion

In conclusion, combining multiple surface science techniques, such as STM, LEED, and ARPES, with ab initio DFT calculations, we develop a consistent picture of the adsorption behavior of Ni-TPP on Cu(100) and clarify the electronic structure of this organic/metal interface. Ni-TPP molecules form two long-range ordered domains, mirrored with respect to the [001] direction and commensurate with the substrate. Van der Waals-corrected DFT calculations reveal the Ni-TPP adsorption geometry: the molecules form a very close-packed arrangement, having the macrocycle lying only about 2 Å above the metal surface. As a consequence, the four phenyl side groups of Ni-TPP tilt and twist upwards, leading to the four main protrusions visible in STM images, in accordance with DFT simulations. Due to the strong interaction of the Cu surface with the macrocycle, pronounced charge rearrangements are observed upon adsorption. In particular, hybrid functional DFT calculations suggest a significant charge transfer from Cu to the molecule resulting in the occupation of the gas-phase LUMO/LUMO + 1 and LUMO + 3 molecular orbitals, accompanied by a back donation of charge from the molecule to the substrate. The interpretation of the three molecular resonances observed in the valence band regime is confirmed by PT results. Thereby, our measured momentum maps allow us to unambiguously assign the photoemission features to molecular states and to accurately determine the azimuthal orientation of the molecule.

We emphasize the importance of complementary STM and $\mu$-ARPES measurements for characterizing such systems. While the former provide information on the molecular states localized on the phenyl rings without being able to probe the porphyrin core, the latter reveal the electronic structure of the frontier orbitals located on the macrocycle. Thus, in general, a multitechnique approach including electronic structure calculations is necessary to develop a consistent picture of the adsorption behavior and electronic properties of interfaces between non-planar molecules and metallic surfaces.

## Methods

**Sample preparation**. The clean surface of Cu(100) was prepared by cycles of $Ar^+$ ion sputtering at 2.0 keV, followed by annealing at 800 K. The surface order and cleanliness were monitored by LEED and photoelectron spectroscopy. A few milligrams of Ni-TPP (Porphyrin Systems) were loaded into a quartz crucible of a home-made Knudsen cell evaporator connected to a separate preparation chamber. Prior to the experiments, the molecules were thoroughly degassed at 480 K for several hours. The molecules were thermally evaporated at 520 K onto the copper substrate kept at room temperature. The growth of the Ni-TPP molecular layer in the photoemission experiments was optimized using LEED. Close to the 1 ML regime, sharp LEED patterns were observed, while additional deposition on an already saturated surface blurred the diffraction spots. This sharp LEED pattern was used as reference for both STM and $\mu$-ARPES experiments. The adsorbed molecular layers were found to be sensitive to beam exposure. In order to minimize radiation damage, spectra were acquired with fast acquisition time at different points of the sample.

**STM**. Low-temperature STM measurements were carried out with an Omicron LT-STM system, at temperature of ~77 K, unless otherwise stated. The microscope is hosted in an ultrahigh vacuum (UHV) chamber, operating at a base pressure of ~$1 \cdot 10^{-10}$ mbar. Images were acquired in constant current mode, with the bias voltage applied to the sample, and the tip at ground. Electrochemically etched tungsten tips were used for imaging. Simulated images have been obtained within the Tersoff-Hamann approximation based on the optimized adsorption structure of Ni-TPP on Cu(100). STM images were processed by subtracting a background plane to correct the sample tilt. The applied enhancements consist of B-spline resampling, to increase the sample count, and mild Gaussian filtering, to remove high-frequency noise components[44].

**μ-ARPES**. The μ-ARPES experiments were performed at the NanoESCA beamline of Elettra, the Italian synchrotron radiation facility, in an UHV system with a base pressure of $5 \cdot 10^{-11}$ mbar, using an electrostatic PEEM[45]. The set-up includes a PEEM column and a double-pass hemispherical analyzer. The instrument, in momentum mode operation, can detect angle-resolved photoemission intensities in the whole-emission hemisphere above the sample surface with extremely high efficiency. The data were collected with 26 eV photon energy, using p-linearly polarized light. The energy and momentum resolution were 75 meV and 0.05 Å$^{-1}$, respectively[46]. The BE scale was referred to the copper Fermi edge. All the photoelectron measurements were performed with the sample at 140 K.

**DFT**. All theoretical photoemission simulations were based on results obtained within the framework of DFT. We have conducted two types of calculations: firstly, for the gas-phase Ni-TPP molecule, which have been performed by the NWCHEM[47] DFT code, using the range-separated HSE hybrid functional[31] for exchange-correlation effects. The simulated momentum maps of the gas-phase Ni-TPP molecule were obtained as the FTs of the respective Kohn–Sham (KS) orbitals[40]. Secondly, we performed calculations for monolayers of Ni-TPP adsorbed on the Cu(100) surface, for which the VASP code[48, 49] was used. We have employed a repeated slab approach, where the metallic substrate was modeled by four metallic layers, and a vacuum layer of ≈15 Å was added between the slabs. To avoid spurious electrical fields, a dipole layer was inserted in the vacuum region[50]. Two types of approximations for exchange-correlation effects have been used: the GGA[51] for the geometry optimization and the HSE functional[31] for a subsequent calculation of the electronic structure. We used a Monkhorst-Pack $3 \times 3 \times 1$ grid of k-points[52] and the projector augmented wave[53] approach, allowing a relatively low kinetic-energy cutoff of ~500 eV. The super cell geometry has been constructed according to the experimental LEED structure shown in the supporting information. During the geometry optimization, the atomic positions of the molecular layer and the first metallic layer were allowed to relax. In order to account for van der Waals interactions, we employed the vdW-surf method according to Ruiz et al.[26, 27] during the geometry optimization.

**Data availability**. The authors declare that relevant data supporting the findings of this study are available on request.

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

## Acknowledgements

All authors thank Margherita Marsili and Paolo Umari for useful discussions on the data interpretation. B.K., D. L. and P.P. acknowledge support from the Austrian Science Fund (FWF), Project P27649-N20 and the computer facilities of the University of Graz and the Vienna Scientific Cluster (VSC).

## Author contributions

G.Z., M.J., and V.F. conceived and performed the angle-resolved photoemission experiment, G.Z analyzed the experimental photoemission data. Z.F. and C.D. carried out the STM measurements and consequent analysis. D.L., P.P., and B.K. performed and analyzed the ab initio DFT calculations. M.P., G.D.S., and A.G. performed preliminary STM experiments. G.Z., D.L., and Z.F. prepared the figures. G.Z. and V.F., with the assistance of D.L., P.P., Z.F., and C.D., drafted the manuscript, which was intensively discussed together with C.M.S. and G.C. All authors discussed the results and reviewed the manuscript.

## Additional information

**Competing interests:** The authors declare no competing financial interests.

**Change History:** A correction to this article has been published and is linked from the HTML version of this article.

