## [Peer Review File · Nature Communications]

Reviewers' comments:

Reviewer #1 (Remarks to the Author):

Prof. Schneider et al. report a multi-method approach regarding the self-assembly and electronic structure of NiTPP on Cu(100). Importantly, an unprecedented filling of the unoccupied orbitals up to LUMO+3 is demonstrated, by the combination of photoemission spectra, micro-ARPES and DFT. These studies will pave new avenues for elucidating the intricate characteristics of the molecular/metal interface, while simultaneously will stimulate further efforts to combine space-average techniques with local probes and theoretical calculations.

The manuscript is very well written, nicely explained, the experiments are well conducted, the simulations reliable, and the results are very sound. Thus, I would strongly support publication provided the following comments are addressed:

- 1.- Major concern. The authors have used a low temperature STM at 77 K. Even though at nitrogen temperature STS is tricky, I would recommend to acquire STS above the macrocycle, in order to reinforce their claims. Furthermore, at 4K access to the electronic structure of the molecule via dI/dV manuscript could be a great addition to the manuscript. Despite the bulky legs, great valuable information is always provided by these powerful local spectroscopies.
- 2.- Major concern. Figure 2a shows two kind of brightness on the molecular species. How many of these dark species were present on the sample? What are they? Adsorption on defects? Alternative conformations? I would recommend to introduce large scale images to check this important issue out.
- 3.- A further emphasis of the extension of their approach to bulky molecules, precluding proper STS, should be introduced in the manuscript.
- 4.- Page 14. Typo "onovera"
- 5.- References are not formatted according to Nature Communications' style

Reviewer #2 (Remarks to the Author):

The interaction of organic molecules with metal substrates and the charge transfer between substrate and molecule is certainly an interesting topic that currently receives a lot of attention. With regard to that, one of the central claims of the authors is:

" The exceptional charge transfer leads to filling of the unoccupied orbitals up to LUMO+3."
(Quotation from abstract).

This statement is not correct. Such charge transfer into adsorbed molecules has been observed before using e.g. NEXAFS spectroscopy and also investigated in quite some detail theoretically (Nature Chemistry, 2010, DOI: 10.1038/NCHEM.591). The observation of charge transfer into an adsorbed molecule thus is not "unexpected".

The authors of the present manuscript need to refer to this and other previous works and then to modify and weaken their claims.

Only after the authors have put their findings into the context of the present state-of-the art, the paper can be evaluated.

In the present form, the paper has to be rejected.

Reviewer #3 (Remarks to the Author):

This paper presents a very interesting combined STM, LEED, ARPES and DFT study of the adsorption of a metalated porphyrin molecule on a metal surface. A new approach based on photo emission tomography is used to determine the occupancy of the molecular frontier orbitals. In particular, they find an unusual electron filling of unoccupied molecular orbitals up to LUMO+3. A similar filling of frontier orbitals was found in DFT calculations of base porphyrin on Cu(110) but was not directly confirmed by the experiments [Ref. 8]. This study is scientifically sound but being a theoretician I am most confident about the DFT part. The conclusions are well-supported.

I have only a few minor comments which needs to be addressed before the paper can be published.

p. 8 , first paragraph: The statement "the phenyl rings remain almost gas-phase-like" is unclear and needs to be substantiated.

p. 10, first paragraph: In TH theory, "the non-zero radius of the tip apex" is accounted for by using the distance between the tip-apex and the surface. An increase of this radius corresponds to an increase of this distance for a fixed distance between the tip and the surface. The distances between the tip-apex and the surface used in the STM simulations should be quoted.

In Fig. 4B, it would be interesting to indicate the positions of the corresponding molecular orbitals of the isolated molecule when aligned with respect to the vacuum level.

Reviewer #1 (Remarks to the Author):

Prof. Schneider et al. report a multi-method approach regarding the self-assembly and electronic structure of NiTPP on Cu(100). Importantly, an unprecedented filling of the unoccupied orbitals up to LUMO+3 is demonstrated, by the combination of photoemission spectra, micro-ARPES and DFT. These studies will pave new avenues for elucidating the intricate characteristics of the molecular/metal interface, while simultaneously will stimulate further efforts to combine space-average techniques with local probes and theoretical calculations.

The manuscript is very well written, nicely explained, the experiments are well conducted, the simulations reliable, and the results are very sound. Thus, I would strongly support publication provided the following comments are addressed:

We thank the referee for the careful reading of the manuscript and for the constructive comments.

Comment 1:

Major concern. The authors have used a low temperature STM at 77 K. Even though at nitrogen temperature STS is tricky, I would recommend to acquire STS above the macrocycle, in order to reinforce their claims. Furthermore, at 4 K access to the electronic structure of the molecule via dI/dV manuscript could be a great addition to the manuscript. Despite the bulky legs, great valuable information is always provided by these powerful local spectroscopies.

Response:

We did not perform systematic STS measurements above the Ni-TPP molecule, as a series of STM images acquired with 0.2 V steps in the [-2.0;+2.0] V range (a selection was shown in the old Fig. S1) did not present any obvious change of contrast on the macrocycle region. This is likely due to the fact that, owing to the strong interaction with the substrate, the macrocycle of Ni-TPP is located very close to the surface, forcing the phenyl ligands to bend upwards. The resulting geometry of such bulky molecules thus prevents us from observing STS features specifically arising from the macrocycle. This is in agreement with the simulated PDOS reported in the manuscript (see Figure 4): indeed, in the [-3.0;+2.0] eV range, the projected DOS on phenyl group has a rather featureless plateau, whereas beyond this range it increases very rapidly. In light of these measurements, which simply confirm the inability of STM to specifically probe the macrocycle states, we have therefore chosen not to further investigate the system by local STM spectroscopy. A statement regarding the STS measurements has been added in the SI. We updated the Fig. S1 as well, by adding the entire sequence of STM images taken at different bias voltages.

In the SI we inserted the following text:

The Ni-TPP appearance in the STM images does not noticeably change for biases in the [-2.0;+1.0] V range, while the phenyl features start to blur when approaching the +2.0 V bias (see Figure S1). **The absence of significant contrast changes on the macrocycle region, when varying the bias voltage in the whole range, suggests that even when the tip is placed above the TPP center, the states related to the macrocycle cannot be deconvolved from the contribution of the phenyl ligands. This is in agreement with the simulated PDOS reported in the manuscript (see Figure 4c). Indeed, in the [-3.0;+2.0] eV range, the projected DOS on phenyl group has a rather featureless plateau, whereas beyond this range it increases very rapidly.**

Figure S1 has been updated in the Supporting Information:

We add the entire sequence of STM images taken at different bias voltages in the [-2.0;+2.0] V range, with a 0.2 V step.

Comment 2:

Major concern. Figure 2a shows two kind of brightness on the molecular species. How many of these dark species were present on the sample? What are they? Adsorption on defects? Alternative conformations? I would recommend to introduce large scale images to check this important issue out.

Response:

These two uncommon features were found a few times also on other STM images taken at different sample positions. In our opinion, they can be related to a chemically modified rectangular-Ni-TPP possibly formed during the deposition [ref. 25, Di Santo *et al.*, Chem. - Eur J., **17**, 14354 (2011)], as well as to impurities, such as H₂TPP, for example. The presence of these species depends on the degassing procedure of the Knudsen cell and on the chosen evaporation temperature. The amount of these species on the surface is very low as confirmed by the quantitative analysis of the core level spectra that do not show additional peaks and are characterized by peak intensity ratios well in agreement with the stoichiometry of the Ni-TPP molecules. For example, the presence of a single peak in the N 1s spectrum suggests that all four nitrogen atoms of the deposited molecules are chemically equivalent, proving that the porphyrin is indeed metalated. Based on this analysis, we can conclude that the amount of the uncommon species/ defects on the surface after NiTPP deposition is below the XPS sensitivity.

In the manuscript we inserted the following text:

The excellent agreement between the measured and simulated LEED patterns of the organic film (see Figure 2b), based on the matrices determined above, confirms the STM findings, also indicating the existence of long-range order across the surface. **Additionally, uncommon features, having a different appearance with respect to the single and close-packed Ni-TPPs, are**

also observed in Fig.2a. These features can be related to species/ defects on the surface, such as chemically modified rectangular-Ni-TPP or H₂TPP, for example [ref. 25, Di Santo, Chem. – Eur. J., **51**, 14354 (2011)]. A quantitative analysis of the C 1s and N 1s core level spectra (not shown), reveals that the peak intensity ratios well agree with the Ni-TPP stoichiometry. Furthermore, no additional features have been detected in the spectra, suggesting that the amount of these species/ defects on the Ni-TPP layer is below the XPS sensitivity.

Comment 3:

A further emphasis of the extension of their approach to bulky molecules, precluding proper STS, should be introduced in the manuscript.

In the conclusions we inserted the following text:

[...] We emphasize the **importance of complementary STM and μ -ARPES measurements for characterizing such systems**. While the former provides information on the molecular states localized on the phenyl rings **without being able to probe the porphyrin core**, the latter reveals the electronic structure of the frontier orbitals located on the macrocycle. Thus, in general, a multi-technique approach including support from electronic structure calculations is necessary to develop a consistent picture of the adsorption behavior and electronic properties of interfaces between non-planar molecules and metallic surfaces.

Comment 4:

Page 14. Typo "onovera"

Response:

the typo has been corrected.

In the manuscript the “onovera” has been substituted with:

on

Comment 5:

References are not formatted according to Nature Communications' style

Response:

from the author guide we understood that the bibliography style will be formatted accordingly to Nat. Comm. prior to publication if the paper will be accepted.

Reviewer #2 (Remarks to the Author):

The interaction of organic molecules with metal substrates and the charge transfer between substrate and molecule is certainly an interesting topic that currently receives a lot of attention.

With regard to that, one of the central claims of the authors is:

"The exceptional charge transfer leads to filling of the unoccupied orbitals up to LUMO+3." (Quotation from abstract). This statement is not correct. Such charge transfer into adsorbed molecules has been observed before using e.g. NEXAFS spectroscopy and also investigated in quite some detail theoretically (Nature Chemistry, 2010, DOI: 10.1038/NCHEM.591). The observation of charge transfer into an adsorbed molecule thus is not "unexpected".

The authors of the present manuscript need to refer to this and other previous works and then to modify and weaken their claims.

Only after the authors have put their findings into the context of the present state-of-the art, the paper can be evaluated.

In the present form, the paper has to be rejected.

Response:

We thank referee for the criticism, however, we think that there has been some misunderstanding in what we meant using the word "unexpected". The main point of the manuscript is not to demonstrate the observation of a charge transfer phenomena that, we agree, has already been observed for different organic/ metal systems using different spectroscopy techniques (including the paper quoted by the referee in his/ her report). Rather, our point is to demonstrate the high and unconventional *magnitude* of such transfer, which takes place at the Ni-TPP/ Cu(100) interface. The filling of the higher unoccupied orbitals up to LUMO+3 is indeed not expected and it was not even considered for similar systems, such as Co-TPP/ Cu(110) [ref. 28, P. Donovan *et al.*, Chem. – Eur. J., **16**, 11641 (2010)].

The photoemission tomography method implemented in our PEEM setup is a state-of-the art technique which can be used to study the adsorption behavior of molecules on the metal substrates. This approach allows us to measure the valence band spectra and, at the same time, record the images of Fourier transform of molecular states. This cannot be done so straightforwardly by adopting other spectroscopy techniques.

We amended the abstract and the introduction in order to explain more clearly the key concept of the manuscript, trying to avoid possible misunderstandings. The title of the paper has been changed as well. We also added in the reference list [ref. 22, T. Tseng, Nat Chem, **2**, 374 (2010)] the paper indicated by the referee and another more specific paper regarding the charge transfer into a porphyrin layer observed by NEXAFS [ref. 23, K. Diller *et al.*, JPCC, **136**, 014705 (2012)].

In the manuscript we made the following changes:

Title: Multi-orbital charge transfer at highly oriented organic/ metal interfaces

Abstract: The molecule-substrate interaction plays a key role in charge injection organic based devices. Charge transfer at molecule-metal interfaces strongly affects the overall physical and magnetic properties of the system, and ultimately the device performance. Here, we report theoretical and experimental evidence of a **pronounced** charge transfer involving nickel tetraphenyl porphyrin (Ni-TPP) molecules adsorbed on Cu(100). The exceptional charge transfer leads to filling of the **higher** unoccupied orbitals up to LUMO+3. As a consequence of the strong interaction with the substrate, the porphyrin's macrocycle sits very close to the surface, forcing the phenyl ligands to bend upwards. Due to this adsorption configuration, scanning tunneling microscopy cannot reliably probe the states related to the macrocycle. We demonstrate that photoemission tomography can instead access the Ni-TPP macrocycle electronic states and determine the reordering and filling of the LUMOs upon adsorption, thereby confirming the remarkable charge transfer predicted by density functional theory calculations.

Introduction: [...] Intense experimental and theoretical efforts have targeted the understanding of the behavior of porphyrin assemblies on metal surfaces [16, 17]. Powerful area-averaging spectroscopic techniques, such as ultraviolet and x-ray photoemission spectroscopy (UPS and XPS) or near-edge x-ray absorption fine structure (NEXAFS) were employed to obtain information on interaction sites [9] and bonding geometries [18], as well as the electronic structure [19] and magnetic properties [20] of adsorbed porphyrin molecular layers. However, these spectroscopic techniques have difficulties to unambiguously identify, and properly describe, the features related to the molecule-metal interaction. For example, interpretation of UPS and NEXAFS spectra, based on the comparison between sub-monolayer and thick (or powder) molecular films, might be misleading. In UPS, the observed structures in the photoemission spectra may arise from purely kinematic factors, such as umklapp scattering [21], rather than molecule-metal hybridization. In NEXAFS, the full occupation of the LUMO leads to the quenching of the related resonances [ref. 22, T. Tseng, Nat Chem, 2, 374 (2010), ref. 23, K. Diller *et al.*, JPCC, 136, 014705 (2012)]. However, the strong polarization dependence of the NEXAFS intensity and charge rearrangement in the molecule, which takes place upon adsorption, may complicate the direct determination of the molecular states involved in the charge transfer process and their energy alignment. Even the calculated density of states can be inaccurate because of the inherent approximations of the theoretical frameworks [24]. For this reason, an experimental technique, providing direct access to the electronic structure of adsorbed molecules, would be desirable to experimentally shed light on the charge redistribution at the molecule-metal interface. In the present work, we unequivocally determine the electronic occupancy of the molecular frontier orbitals, *i.e.* HOMOs and LUMOs, of the

adsorbed nickel tetraphenyl porphyrin (Ni-TPP) on the Cu(100) surface by means of photoemission tomography (PT). Our findings indicate a **pronounced** electron charge transfer from the Cu(100) surface to the macrocycle of the Ni-TPP molecules, involving even the gas-phase LUMO+3 of Ni-TPP. By comparing STM and DFT-simulated images, we demonstrate that the topography contrast arises mainly from the electronic states of the porphyrin phenyl rings, which are strongly tilted upwards. This molecular configuration allows the macrocycle to get close to the copper surface, while preventing the frontier orbitals, spatially located on this moiety, to be probed by STM.

Reviewer #3 (Remarks to the Author):

This paper presents a very interesting combined STM, LEED, ARPES and DFT study of the adsorption of a metalated porphyrin molecule on a metal surface. A new approach based on photo emission tomography is used to determine the occupancy of the molecular frontier orbitals. In particular, they find an unusual electron filling of unoccupied molecular orbitals up to LUMO+3. A similar filling of frontier orbitals was found in DFT calculations of base porphyrin on Cu(110) but was not directly confirmed by the experiments [Ref. 8]. This study is scientifically sound but being a theoretician I am most confident about the DFT part. The conclusions are well-supported.

I have only a few minor comments which needs to be addressed before the paper can be published.

We thank the referee for the positive opinion of our work and for the constructive comments.

Comment 1:

p. 8 , first paragraph: The statement “the phenyl rings remain almost gas-phase-like” is unclear and needs to be substantiated.

Response:

We clarified the sentence in the main text.

In the manuscript we have inserted the following text:

In the former minimum, **the phenyl ligands assume an almost gas-phase conformation ($\theta = 70^\circ$, $\phi = 180^\circ$)**, whereas in the latter they are strongly tilted, as described above.

Comment 2:

p. 10, first paragraph: In TH theory, “the non-zero radius of the tip apex” is accounted for by using the distance between the tip-apex and the surface. An increase of this radius corresponds to an increase of this distance for a fixed distance between the tip and the surface. The distances between the tip-apex and the surface used in the STM simulations should be quoted.

Response:

We quoted the tip apex-copper surface distance in the text.

In the manuscript we have inserted the following text:

However, **the simulated images unavoidably show many features that the STM tip cannot resolve experimentally, due to the broadening caused by the intrinsically large width and the electronic properties of the real tip apex (in the calculations, apex-copper surface distance ~ 5.5 -**

7.5 Å). Therefore, the phenyls in the simulation appear slightly different from the experimental image.

Comment 3:

In Fig. 4B, it would be interesting to indicate the positions of the corresponding molecular orbitals of the isolated molecule when aligned with respect to the vacuum level.

Response:

The position of the molecular orbitals of the isolated molecule has been indicated in the top part of Figure 4b. The caption has been changed accordingly.

In the manuscript we modified the caption of figure 4b:

(a) Photoemission spectra of clean Cu(100) and Ni-TPP/ Cu(100) acquired at 26 eV of photon energy. (b) PDOS of adsorbed molecules onto molecular orbitals for the Ni-TPP/ Cu(100) system. **The energy position of the corresponding gas-phase molecular orbitals, aligned with respect to the vacuum level, is indicated with colored bars on the top axis.** (c) DOS of the entire molecule (Black curve), projected density of states (PDOS) onto the phenyl group (light blue curve) and on macrocycle (green curve).

REVIEWERS' COMMENTS:

Reviewer #1 (Remarks to the Author):

The authors have addressed all points raised by referees in a detailed and convincing way. Thus, I would recommend publication as it is.

Reviewer #2 (Remarks to the Author):

I am happy with the reply of the authors as well as with the changes they did to the manuscript.

I have also studied the comments of the other referees, as well as the corresponding reply by the authors. My impression is that also the concerns of the other referees were carefully considered.

I recommend the modified version of the paper for publication in Nature Communications.

Reviewer #3 (Remarks to the Author):

I have now gone through the response by the authors to the comments and criticisms raised by the three referees. I find that the authors have responded satisfactorily to these comments and criticisms and the paper can now be published as it is.